# Homologous and Heterologous Vaccination Regimens with mRNA and rVSV Platforms Induce Potent Immune Responses Against SFTSV Glycoprotein

**DOI:** 10.3390/v17081095

**Published:** 2025-08-08

**Authors:** Tomaz B. Manzoni, Jonna B. Westover, Kendall A. Lundgreen, Philip D. Hicks, Raegan J. Petch, Jordan T. Ort, Drew Weissman, Steven H. Y. Fan, Scott E. Hensley, Norbert Pardi, Brian B. Gowen, Paul Bates

**Affiliations:** 1Department of Microbiology, Perelman School of Medicine, University of Pennsylvania, Philadelphia, PA 19104, USA; manzonitb@gmail.com (T.B.M.); kendall.lundgreen@pennmedicine.upenn.edu (K.A.L.); hicksp@udel.edu (P.D.H.); rpetch@vet.upenn.edu (R.J.P.); jordan.ort@pennmedicine.upenn.edu (J.T.O.); hensley@pennmedicine.upenn.edu (S.E.H.); pnorbert@pennmedicine.upenn.edu (N.P.); 2Department of Animal, Dairy and Veterinary Sciences, Utah State University, Logan, UT 84322, USA; jonna.westover@usu.edu (J.B.W.); brian.gowen@usu.edu (B.B.G.); 3Institute for RNA Innovation, University of Pennsylvania, Philadelphia, PA 19104, USA; dreww@pennmedicine.upenn.edu; 4Acuitas Therapeutics, Vancouver, BC V6T 1Z3, Canada; sfan@acuitastx.com

**Keywords:** severe fever with thrombocytopenia syndrome virus, *Dabie bandavirus*, mRNA, vesicular stomatitis virus, rVSV, vaccine, immunogenicity

## Abstract

Background: Severe fever with thrombocytopenia syndrome virus (SFTSV) is a highly pathogenic bunyavirus with a high case-fatality ratio for which there is no approved vaccine. Studies have assessed different vaccine technologies. However, few studies have yet assessed the immunogenicity of heterologous prime-boost regimens. Methods: Here, we compare a lipid nanoparticle (LNP)-encapsulated nucleoside-modified mRNA-based vaccine encoding the SFTSV glycoproteins, Gn and Gc, to our recently described recombinant VSV SFTSV (rVSV-SFTSV) vaccine in single dose, homologous, and heterologous prime-boost regimens in mice. Results: We show that all regimens protect from pathogenic SFTSV challenge and elicit strong long-lasting antibody responses. Furthermore, strong cellular immunity is elicited by mRNA-LNP immunizations and by heterologous immunization with an rVSV-SFTSV prime and mRNA-LNP boost. Cellular responses robustly polarized towards a type 1 response, characterized by high levels of IFNγ, TNFα, and IL-2. Immunization with mRNA led to a mixed type 1/type 2 immune response, as determined by antibody isotypes IgG1 and IgG2c. We found that homologous immunization leads to stronger antibody responses while heterologous immunization drives a slightly stronger cellular response. Conclusions: Taken together, the vaccine platforms described here represent strong vaccine candidates for further development.

## 1. Introduction

Severe fever with thrombocytopenia syndrome virus (SFTSV, recently renamed *Dabie bandavirus*, and formerly *Huaiyangshan banyangvirus*) is an emerging tickborne bunyavirus. First identified in China in a 2011 study, SFTSV has since been identified in and is considered endemic to several southeast Asian countries and regions including Japan, South Korea, Vietnam, Taiwan, Pakistan, Myanmar, and Thailand [1,2,3,4,5,6,7,8,9,10,11]. Infection with SFTSV is often clinically mild with symptoms including leukopenia, thrombocytopenia, fever, and the presence of gastrointestinal symptoms [12,13,14]. In more severe cases, hemorrhage and multiorgan dysfunction can occur with patients generally succumbing to multiorgan failure in fatal cases [14]. Initial case fatality ratios (CFRs) reported were approximately 30%, with more recent studies reporting an overall CFR of 8% [15]. The wide distribution of SFTSV coincides with the range of its primary vector, the tick *Haemophysalis longicornis*, though other tick species have since been shown to be competent vectors [16,17,18,19]. The geographic distribution of *H. longicornis* has been greatly expanding and now includes Australia, New Zealand, and the United States, which presents the opportunity for further spread of SFTSV [20,21,22,23,24]. Additionally, though less common, SFTSV has been found to be able to spread through zoonotic and human-to-human pathways through contact with blood, feces, urine, or saliva. Multiple nosocomial outbreaks have been reported, as well as likely transmission from infected domesticated or peri-domesticated cats [25,26,27,28,29,30,31,32,33]. Together, the high CFR and potential for rapid spread of SFTSV along with a lack of approved therapeutics and vaccines have led to the categorization of SFTSV as a high priority pathogen by both the National Institute of Allergy and Infectious Diseases (NIAID) and the World Health Organization (WHO) [34,35].

SFTSV is in the class *Bunyaviracetes* and the family *Phenuiviridae* [36]. Like other viruses in this family, SFTSV is a tri-segmented, single-stranded RNA virus with a negative and ambisense genome. The three segments of the virus are named by their sizes with small (S), medium (M), and large (L) segments [1]. S is ambisense and encodes the nucleoprotein (N) and non-structural protein (NSs), which antagonizes interferon (IFN) [37]. The M and L segments are both negative sense. L encodes the RNA-dependent RNA polymerase (RdRp), which is essential for virus replication. The M segment encodes the envelope glycoprotein, which is translated as a polyprotein and is proteolytically cleaved into its two subunits Gn and Gc [38]. The Gn/Gc complex is the sole virally encoded surface protein on SFTSV virions. Gn mediates receptor binding, and entry occurs via the fusion peptide found in the Gc subunit [39,40,41,42]. The receptors and entry factors for SFTSV remain poorly defined, though multiple host protein attachment and entry factors such as DC-SIGN, UGCG, non-muscle myosin heavy chain IIA, and CCR2 have been identified [42,43,44,45,46]. Critically, studies have shown that antibodies directed against Gn or Gc are able to inhibit viral entry into cells making the Gn/Gc complex the primary target for vaccine development [47,48,49,50,51].

Currently there are no approved vaccines against SFTSV. In part, the lack of vaccines and therapeutics can be attributed to the limitations of animal models. Infection with SFTSV does not cause severe disease in wild-type mice with fully functional immune systems [52,53,54]. Currently, the only immunocompetent animal models for lethal SFTSV disease are cats and aged ferrets [55,56]. The use of these animals for immunologic studies is limited due to the lack of tools for analysis of immune responses and difficulty in handling. *Stat2*^−/−^ hamsters and *Ifnar*^−/−^ mice have been the primary small animal models used for vaccine development against SFTSV due to reliable fatal SFTSV infections, though these models are limited in their translation to immunocompetent humans [52,53,57]. More recently, one study has shown that anti-IFNAR (α-IFNAR) antibodies in wild-type mice confers susceptibility to pathogenic SFTSV challenge, which is characterized by severe weight loss, high SFTSV titers particularly in the spleen and serum, and death. These factors make this a promising model for SFTSV vaccine development as immunocompetent mice can be vaccinated for SFTSV, allowed to develop normal immune responses, and then challenged with a pathogenic dose of SFTSV to evaluate protection [58].

Previous vaccine development against SFTSV has included platforms such as DNA, protein subunit, whole inactivated virus, live attenuated virus, virus-vectored, and mRNA vaccines [59,60,61,62,63,64,65,66,67,68,69,70,71,72,73]. Of the vaccine candidates tested thus far, recombinant vesicular stomatitis virus (rVSV) and mRNA vaccines show the greatest promise for prevention of SFTSV infection and disease due to the positive safety profile exhibited, the potent immunogenicity induced in animal models, and the ease of production and administration [73]. In this study, we directly compare rVSV and mRNA SFTSV vaccines in single dose, homologous boost, and heterologous boost vaccination strategies.

We and others have recently described an rVSV vaccine encoding SFTSV Gn/Gc (rVSV-SFTSV) that was safe and highly effective in preventing disease in *Ifnar*^−/−^ mice [68,69,74]. Vesicular stomatitis virus (VSV) is a livestock pathogen with very low pathogenicity and seroprevalence in the human population [75,76]. VSV is a genetically tractable model, and it is able to readily incorporate foreign glycoproteins onto the surface of its virions to produce recombinant VSV (rVSV). These rVSVs can induce protective humoral and cellular responses, and they have been considered as a vaccine platform for a wide variety of pathogens [75,77,78,79,80,81,82]. Currently, an Ebola virus (EBOV) vaccine using an rVSV platform is approved for use by the United States Food and Drug Administration (FDA), and this vaccine has been distributed to combat recent EBOV outbreaks in Africa with incredible success [79,83,84]. As a virus platform, rVSV-based vaccines elicit strong type 1 immune responses, which are ideal for the elimination of intracellular pathogens such as viruses [85,86,87,88]. A shortcoming of many other vaccine platforms is the skewing of immune responses to a type 2 response, which is more suited to attack extracellular pathogens [85]. Additionally, immune responses to rVSV vaccines are long lived, and studies have reported durable T cell and B cell responses up to 5 years after vaccination with rVSV-EBOV [89,90]. Critically, immunity from rVSV vaccines arises rapidly, with studies showing some protection as early as three days post vaccination [91,92,93]. rVSV vaccines have even been shown to be protective if given post-exposure in some contexts, making them an attractive vaccine platform for tickborne viruses for which possible transmission events can be observed [94,95,96,97]. However, our data from a single immunization in wild-type C57BL/6 mice contradicted a previous report and suggested that rVSV-SFTSV may be insufficient as a single dose vaccine and may require a prime-boost regiment to induce quality, high-titer serological responses in immunocompetent animals [69].

Many of the most recently developed SFTSV vaccines utilize the FDA-approved vaccine platform messenger RNA (mRNA). mRNA vaccines use an in vitro transcribed mRNA strand, which includes modified nucleosides and is encapsulated in a lipid nanoparticle, to elicit production of an antigen in the vaccinated individual. This platform is powerful, allowing rapid design and vaccine manufacturing pipelines for emerging pathogens [98]. Recent vaccination campaigns against SARS-CoV-2 have demonstrated the effectiveness of mRNA vaccines in controlling viral infections. However, reports from the COVID-19 pandemic have largely indicated that three or more doses of the SARS-CoV-2 mRNA vaccine are required for complete protection of people [99]. Studies on multi-dose mRNA SFTSV vaccines have shown remarkable success in animal models [61,62,63,100] but the efficacy of a single dose of mRNA SFTSV vaccines has not been reported. Overall, the mRNA platform represents a potential strong candidate for future vaccine development but remains understudied in its use against SFTSV.

In light of the severe acute respiratory syndrome coronavirus 2 (SARS-CoV-2) pandemic, heterologous vaccination strategies have come to the forefront of discussion in the scientific community. Heterologous vaccination strategies may improve host protection by activating specific immune pathways tailored to the specific target. One report using an adenovirus-vectored prime followed by an mRNA boost demonstrated an increased antibody titer after completion of the regimen compared to titers achieved by an mRNA prime followed by an adenovirus-vectored vaccine boost [101,102,103,104]. Regarding cellular immunity, current data indicates heterologous vaccination induces similar or better responses compared to homologous vaccination [101,102,103,104]. Currently, detailed analysis of immunologic phenotypes upon heterologous vaccination remains largely undefined and attempts to analyze these responses are predominantly limited to SARS-CoV-2 vaccines. Here, we utilize a novel mouse model using wild-type C57BL/6 mice and α-IFNAR antibody blockade to compare a full-length SFTSV Gn/Gc mRNA vaccine to our previously described rVSV-SFTSV vaccine in single dose as well as prime-boost studies. Additionally, we consider a heterologous prime-boost strategy and show that all vaccines tested elicit powerful antibody responses. Immunization with homologous mRNA and heterologous vaccination yield comparable antibody and type 1 cytokine responses in T-cells with heterologous immunization being slightly superior in eliciting CD4+ responses and degranulation of CD8+ T-cells. Furthermore, all immunization strategies protect mice from pathogenic SFTSV challenge. Overall, our analysis suggests the mRNA platform is a strong candidate for development of a human SFTSV vaccine with homologous and heterologous prime-boost regimens inducing potent immune responses.

## 2. Materials and Methods

### 2.1. Ethics Statement

All animals were treated ethically, complying with guidelines set by the United States Department of Agriculture (USDA) and the Utah State University Institutional Animal Care and Use Committee and the University of Pennsylvania Laboratory Animal Resources guidelines. During the challenge studies, animals were humanely euthanized when they lost weight equal to or greater than 30% of their body weight, according to approved endpoints.

### 2.2. Cells, Viruses, and Mice

ATCC verified and mycoplasma free HEK293T and Vero E6 cells were maintained in DMEM (Gibco, Waltham, MA, USA #11965084) containing 10% cosmic calf fortified serum (Cytiva, Logan, UT, USA, #SH30087.03), 2mM L-glutamine (Corning, Tewksbury, MA, USA, #25-005-Cl), and 1mM Sodium Pyruvate (Gibco, Waltham, MA, USA, #11360-070). Cells were passaged every 2–3 days and maintained for no more than 20 passages.

Recombinant viruses encoding heterologous viral glycoproteins in genomic position 4 (rVSV-SFTSV and rVSV-EBOV) were launched and described previously [46,69,105]. rVSV-SFTSV and rVSV-EBOV also contain an additional open reading frame in position five encoding mCherry. All recombinant viruses were grown in Vero E6 cells by infecting a confluent T-175 flask at an MOI of 0.3–0.5. Virus was collected at 48–72 h post infection with the addition of Hepes buffer pH 7.4 to 25 mM. Media was clarified by centrifuging at 6000× *g* for 5 min at 4 °C twice. Virus was then frozen at −80 °C until used for ultracentrifugation. Virus was concentrated by ultracentrifuging virus-containing media through a 20% sucrose gradient at 115,500× *g* for 2 h at 4 °C using SW-32 tubes in a Beckman Coulter Optima XPN-80 ultracentrifuge. After removal of the sucrose and media, pelleted virus was placed on ice with 500 µL Hepes buffered saline overnight. The next day, virus pellets were resuspended and frozen at −80 °C. Viral titer was determined by plaque assays on Vero E6 cells with a 1.25% Avicel RC-591 NF (DuPont, Newark, DE, USA, #RC591-NFBA500) overlay and then stained with 1% crystal violet [106]. As previously described, sequence analysis of the rVSV-SFTSV discovered that the virus acquired two nonsynonymous mutations that caused the substitutions E982K and K1071E within Gc [69].

SFTSV, strain HB29, was obtained from Dr. Robert Tesh (WRCEVA; World Reference Center for Emerging Viruses and Arboviruses at the University of Texas Medical Branch, Galveston, TX, USA). The virus stock (5.6 × 10^6^ PFU/mL; 1 passage in Vero E6 cells) used was from a clarified cell culture lysate preparation. Virus stock was diluted in sterile minimal essential medium (MEM) and inoculated by intraperitoneal (IP) injection of 0.2 mL containing approximately 26, 260, 2600, or 26,000 CCID_50_ for LD_50_ studies, and 26,000 CCID_50_ in challenge studies. For challenge experiments, animals were given 1 mg of α-IFNAR1 monoclonal antibody clone MAR1-5A3 (Leinco Technologies, St. Louis, MO, USA, #I-1188) by IP injection one day pre-challenge (−1) and 0.5 mg of α-IFNAR1 monoclonal antibody on day 2 post challenge.

C57BL/6 mice were ordered from Jackson Labs (Bar Harbor, ME, USA). All mouse experiments were performed using equal numbers of male and female mice. All mice were given approximately 7 days to acclimate to their cages and vivarium prior to each experiment. Eight-week-old mice were used to calculate the lethal dose of SFTSV (Figure 1) and evaluate the antibody and T cell responses prior to challenge (Figure 2, Figure 3 and Figure 4). Four-week-old C57BL/6 mice were used for the SFTSV challenge studies (Figure 5). Mice were weighed immediately prior to all vaccination and infection procedures. Mice receiving intradermal injections at the University of Pennsylvania were anesthetized using 1% isoflurane in air delivered by vaporizer (Northern Vaporisers, Skipton, UK) to the anesthesia chamber. Mice at Utah State University were anesthetized for intradermal injections using a bell jar and open-drop method. Injection sites were first prepared by cleaning with a 70% ethanol pad. Vaccination experiments without authentic SFTSV challenge were performed under animal biosafety level (ABSL) 2 conditions at the University of Pennsylvania. All other vaccination experiments that included authentic SFTSV challenge were performed in ABSL3 conditions at Utah State University.

### 2.3. Production of mRNA-LNP Vaccines

The codon-optimized Gn/Gc glycoprotein gene from SFTSV and firefly luciferase were synthesized (Genscript, Piscataway, NJ, USA) and cloned into an mRNA production plasmid. A MEGAscript T7-driven in vitro transcription reaction (Ambion, Austin, TX, USA, #AM1334) using linearized plasmid template was performed to generate mRNA with 101 nucleotide long poly(A) tail. One-methylpseudouridine (m1Ψ)-5′-triphosphate (TriLink Biotechnologies, San Diego, CA, USA, #N-1081) instead of UTP was used to generate modified nucleoside-containing mRNA. RNAs were capped using the ScriptCap m^7^G capping kit with 2′-O-methyltransferase (CellScript, Madison, WI, USA, #C-SCCE0625 to obtain cap1 as described [107]. Cellulose-based purification of mRNAs was performed as described [108]. All mRNAs were then tested on an agarose gel before storing at −20 °C. The cellulose-purified m1Ψ-containing mRNAs were encapsulated in LNP using a self-assembly process as previously described wherein an ethanolic lipid mixture of ionizable cationic lipid, phosphatidylcholine, cholesterol and polyethylene glycol-lipid was rapidly mixed with an aqueous solution containing mRNA at acidic pH [109]. The RNA-loaded particles were characterized and subsequently stored at −80 °C at a concentration of 1 mg/mL.

### 2.4. Enzyme-Linked Immunosorbent Assay (ELISA)

Production of SFTSV Gn and Gc ectodomains: The SFTSV Gn ectodomain (amino acids 20-452) and Gc ectodomain (amino acids 562-996) were cloned from pCAG-SFTSV Gn/Gc and put into the pHLsec expression vector which adds a secretion signal and C-terminal His tag to the inserted sequence. FreeStyle 293-F cells grown in FreeStyle 293 Expression Media (Gibco, Waltham, MA, USA, #12338018) were co-transfected using 293Fectin (Gibco, Waltham, MA, USA, #12347019) with pHLSec-SFTSV Gn and pHLSec-SFTSV Gc plasmids encoding ectodomains of SFTSV Gn or Gc. Supernatants containing SFTSV Gn and Gc were collected 4 days post-transfection and purified by nickel-nitrilotriacetic acid resin (Qiagen, Hilden, Germany, #30210) according to the manufacturers protocol. Eluted protein was concentrated and buffer exchanged into PBS using an Amicon Ultra-15 Centrifugal Filter with a 30 kDa MWCO (Millipore Sigma, Burlington, MA, USA, #UFC903024) and frozen at −80 °C.

ELISA: Immulon 2HB (Thermo Fisher Scientific, Waltham, MA, USA, #3455) plates were coated with 0.75 µg/mL of purified SFTSV Gn/Gc ectodomains in sodium carbonate at 4 °C overnight. The next day, ELISA plates were washed with phosphate-buffered saline containing 0.1% Tween-20 (PBS-T) and blocked for an hour at room temperature with 3% milk in PBS-T. Mouse sera was diluted in 1% milk in PBS-T and serially diluted 2-fold with an initial dilution of 1:50, 1:800, or 1:3200. Plates were incubated with diluted mouse sera for 2 h at room temperature. Secondary HRP conjugated antibodies for total IgG (GE Healthcare, Chicago, IL, USA, #NA931), IgG1 (Cell Signaling Technology, Danvers, MA, USA, #96714S), and IgG2c (Cell Signaling Technology, Danvers, MA, USA, #56970S) were diluted in 1% PBS-T at 1:5000, 1:6000, or 1:4000, respectively, and staining was performed at room temperature for an hour. Prior to experiments, IgG1 and IgG2c antibodies were normalized against known concentrations of their target antigens so comparisons between isotypes could be made. SureBlue TMB 1 component substrate (KPL, Gaithersburg, MD, USA, #52-00-01) was then added to plates and quenched after 5 min with 250 mM HCl. Absorbance at 450 nm was immediately read on a SpectraMax 190 microplate reader (Molecular Devices, San Jose, CA, USA). Endpoints were determined as OD values twice as high as the background on a negative control run on each plate. Samples lacking absorbance at our lowest dilution of 1:50 were assigned a titer of 25 signifying titers below the limit of detection.

### 2.5. Immunizations

Vaccines were diluted to the desired concentrations with sterile PBS immediately prior to vaccination. Vaccination was performed via IP injection of 200 µL inoculum for rVSV vaccines and intradermal injection of 50 µL inoculum for mRNA vaccines.

### 2.6. Blood Collection

Mice were anesthetized with 1% isoflurane as described above. Blood was collected through the submandibular route using Goldenrod lancets 5 mm (Medipoint, Mineola, NY, USA) and maintained on ice after collection. Serum was separated from blood by centrifugation at 8000 RPM for 30 min at 4 °C in an Eppendorf 5424R centrifuge (Eppendorf, Enfield, CT, USA). Serum was heat inactivated by incubating at 56 °C for 30 min. Serum was frozen at −80 °C for long-term storage. After thawing, serum was stored at 4 °C for short-term (less than 7 days) to complete ELISA and neutralization assay replicates.

### 2.7. Pseudovirus Neutralization Assay

Production of VSV pseudotype with SFTSV Gn/Gc: HEK293T cells plated 24 h previously at 2 × 10^7^ cells per T-175 flask were transfected with 45 μg of pCAG-SFTSV Gn/Gc expression plasmid using Lipofectamine 2000 (Invitrogen, Waltham, MA, USA, #11668-019) according to the manufacturer’s protocol. Thirty hours after transfection, the SFTSV Gn/Gc expressing cells were infected for 2–4 h with VSV-G pseudotyped VSVΔG-mNeon at an MOI of ~1–3. The VSV-G pseudotyped VSVΔG-mNeon was generated by deleting the cognate VSV-G and linking mNeon to the N-terminus of P, and the virus was launched as previously described [46]). After infection, the cells were washed twice with FBS-free media to remove unbound virus. Media containing the VSVΔG-mNeon SFTSV Gn/Gc pseudotypes was harvested 30 h after infection and clarified by centrifugation twice at 6000× *g* then aliquoted and stored at −80 °C until used for antibody neutralization analysis.

Antibody neutralization assay using VSVΔG-mNeon SFTSV Gn/Gc: Vero E6 cells were seeded in 100 μL at 2 × 10^4^ cells/well in a 96-well collagen coated plate. The next day, 2-fold serially diluted serum samples were mixed with VSVΔG-mNeon SFTSV Gn/Gc pseudotype virus (100–200 focus forming units/well) and incubated for 1 h at 37 °C. Also included in this mixture to neutralize any potential VSV-G carryover virus was 1E9F9, a mouse anti-VSV Indiana G, at a concentration of 600 ng/mL (Millipore Sigma, Burlington, MA, USA, #MABF2321). The antibody-virus mixture was then used to replace the media on VeroE6 cells. 16 h post infection, the cells were washed and fixed with 4% paraformaldehyde before visualization on an S6 FluoroSpot Analyzer (CTL, Shaker Heights, OH, USA). Individual infected foci were enumerated, and the values compared to control wells without serum. The focus reduction neutralization titer 50% (FRNT_50_) was measured as the greatest serum dilution at which focus count was reduced by at least 50% relative to control cells that were infected with pseudotype virus in the absence of mouse serum. FRNT_50_ titers for each sample were measured in two to three technical replicates performed on separate days. The limit of detection of the FRNT_50_ assay is a reciprocal dilution of 20.

### 2.8. Flow Cytometry

Mouse spleens were harvested and placed in RPMI (Gibco, Waltham, MA, USA, #11875-085) with 10% FBS (Corning, Tewksbury, MA, USA, #35-010-CV). Spleens were smashed between two slides to make single cell suspensions. Cells were filtered through a 70 µm cell strainer (BioLogix, Saint Louis, MO, USA, #15-1070). ACK lysis buffer (Quality Biological, Gaithersburg, MD, USA, Cat #118-156-101) was used to lyse red blood cells. Remaining cells were then resuspended in RPMI containing 10% FBS, glutamine (Corning, Tewksbury, MA, USA, #25-005-Cl), and β-mercaptoethanol (Millipore Sigma, Burlington, MA, USA, #M7522). Peptide pools containing SFTSV Gn or SFTSV Gc peptides (GenScript, Piscataway, NJ, USA) were added to cells at 1.5 µg/mL for an hour at 37 °C in the presence of α-CD28 antibody (BD Biosciences, Franklin Lakes, NJ, USA, #553295). GolgiPlug (BD Biosciences, Franklin Lakes, NJ, USA, #555029) and GolgiStop (BD Biosciences, Franklin Lakes, NJ, USA, #554724) were then added to cells and incubated overnight at 37 °C. If staining for CD107a PE-Cy7, (BD Biosciences, Franklin Lakes, NJ, USA, #560647) antibody was added along with GolgiPlug/GolgiStop and stained overnight. Extracellular staining was then performed in FACS buffer for 30 min at 4 °C with L/D Aqua (Invitrogen, Waltham, MA, USA, #L34957), α-CD8 Pacific Blue (Biolegend, San Diego, CA, USA, #100725), and α-CD4 PerCP/Cy5.5 (Biolegend, San Diego, CA, USA, #100434). Cells were fixed using BD fix/perm solution (BD Biosciences, Franklin Lakes, NJ, USA, #554722) then stained intracellularly with α-IL-2 BV711 (Biolegend, San Diego, CA, USA, #503837), α-TNFα PE-Cy7 (BD Biosciences, Franklin Lakes, NJ, USA, #557644), α-IFNγ AF-700 (BD Biosciences, Franklin Lakes, NJ, USA, Ca#557998), α-CD3 APC-Cy7 (BD Biosciences, Franklin Lakes, NJ, USA, #557596), and α-Granzyme B (BD Biosciences, Franklin Lakes, NJ, USA, #560213). Cells were analyzed on a BD LSRII flow cytometer with high-throughput system using FACSDIVA software version 9.0 (BD Biosciences, Franklin Lakes, NJ, USA). Flow data was then analyzed and prepared for presentation with FlowJo software v10.10.0 (FlowJo LLC, Ashland, OR, USA).

### 2.9. Serum and Tissue Virus Titers

Virus titers were assayed using an infectious cell culture assay as previously described [110]. Briefly, a specific volume of tissue homogenate or serum was serially diluted and added to quadruplicate wells of Vero E6 cell monolayers in 96-well microtiter plates. The viral cytopathic effect (CPE) was determined 10 days after plating and the 50% endpoints were calculated as described [111]. The assay’s lower limits of detection were 1.67 log_10_ CCID_50_/mL serum and 2.27 log_10_ CCID_50_/g tissue.

### 2.10. Statistical and Data Analysis

All serological assays were analyzed with two-way analysis of variance (ANOVA) with Tukey’s multiple comparisons test. T-cell assays were analyzed with ordinary one-way ANOVAs with Tukey’s multiple comparisons test. All statistical evaluations were completed on Prism 9 (GraphPad Software, La Jolla, CA, USA).

## 3. Results

### 3.1. IFNAR Antibody Blockade Recapitulates Pathogenic SFTSV Infection in WT Mice

Previous studies using various viruses have shown that using an α-IFNAR monoclonal antibody blockade can lead wild-type mice to succumb to infections that are generally non-lethal [112]. Indeed, a previous study using a different strain of SFTSV (KH1 strain) showed that wild-type C57BL/6 mice treated with an α-IFNAR monoclonal antibody succumbed to SFTSV infection with similar, but delayed, symptoms and disease manifestations as seen in the more commonly used *Ifnar*^−/−^ mouse model. Since the *Ifnar*^−/−^ model cannot mount a comprehensive immune response upon vaccination, we optimized an α-IFNAR blockade model with SFTSV strain HB29. This allows us to fully characterize immune responses to candidate vaccines in immunocompetent animals and then challenge them with pathogenic SFTSV doses [113,114]. Animals were given two doses of the α-IFNAR monoclonal antibody, one dose prior to challenge (day −1) and one dose on day 2 post infection (Figure 1A). Several challenge doses were used ranging from 26 to 26,000 cell culture infectious dose 50% (CCID_50_) to determine the median lethal dose (LD_50_). Six 8-week-old C57BL/6 mice were included in each dose group. Challenge with SFTSV caused severe weight loss with animals beginning to succumb at 6 days post infection (dpi) (Figure 1B,C). By day 9, all animals challenged with 26,000 CCID_50_ succumbed (Figure 1C). Interestingly, the lowest challenge dose (26 CCID_50_) achieved the second highest mortality with 50% of animals succumbing to infection (Figure 1C). From these data, the LD_50_ was determined to be 18,000 CCID_50_. Based on these results, a challenge dose of 26,000 CCID_50_ was selected for future viral challenges.

### 3.2. Prime-Boost Vaccination Strategies Induce Powerful Antibody Responses Against SFTSV

To assess the efficacy and immunogenicity of an SFTSV mRNA vaccine relative to the previously described rVSV-SFTSV, we immunized wild-type C57BL/6 mice in either single dose or prime-boost regimens (Figure 2A). Single dose vaccine conditions included SFTSV Gn/Gc mRNA (RS), rVSV-SFTSV (VS), and as negative controls, firefly luciferase mRNA (RL), and rVSV-EBOV (VE). Prime-boost regimens included mRNA homologous regimens RS+RS and the negative control RL+RL, viral homologous regimen VS+VS, the heterologous regimen VS+RS, and the negative control VE+RL (Figure 2A). A heterologous vaccination strategy using an mRNA prime and rVSV boost was not considered due to rVSV-SFTSV containing SFTSV Gn/Gc on the virion surface which would likely be neutralized by antibodies elicited by an mRNA vaccination, thereby preventing any significant boost. Six 8-week-old C57BL/6 mice were included in each vaccination group. Singly vaccinated animals were euthanized at 7 days post immunization for T-cell analysis and at 14 days for antibody assays (Figure 2A). For boosted animals, a booster dose was administered at 21 days post prime, and animals were euthanized at day 28 post initial immunization for T-cell analysis and day 35 for FRNT_50_ and ELISA antibody assays (Figure 2A). Immunization did not cause significant weight loss regardless of vaccination regimen (Figure 2B). All mice showed a slight drop in weight the day after immunization, likely from handling stress. All mice recovered by 2 days post immunization (Figure 2B).

Using an SFTSV Gn/Gc ectodomain enzyme-linked immunosorbent assay (ELISA), we determined total IgG titers induced by each vaccination strategy. In single dose vaccinated mice, the total IgG endpoint titers reached were identical in both RS and VS immunized mice (Figure 2C). In sera from 21 days post immunization, mice primed with RS and VS continued to have comparable endpoint titers, though these titers are approximately 2-fold higher than on day 14 (Figure 2C). After boost, homologous RS+RS and heterologous VS+RS vaccinated mice had similar endpoint titers with geometric means of approximately 129,000 and 100,000, respectively (Figure 2C). Homologous VS+VS immunized mice had the lowest titers at approximately 32,000 (Figure 2C). Fold changes in total IgG titers upon boost had a similar 15- and 14-fold titer increase upon boost between RS+RS and VS+RS groups, respectively. Mice immunized with the VS+VS regimen had a 3-fold increase in total IgG titers upon boost.

We next sought to characterize the neutralization ability of vaccinated serum. At 14 days post vaccination, all single dose vaccinated mice had sero-converted with geometric mean neutralizing titers of 500 and 170 for RS and VS, respectively (Figure 2D). Prime-boost vaccination regimens induced powerful neutralizing antibody titers from all groups. At 21 days post vaccination, serum titers were slightly increased from what was observed at day 14 in singly vaccinated mice (Figure 2D). After boosting, all mice had increased neutralizing antibody titers (Figure 2D). Homologous RS+RS immunized mice had higher titers than all other groups with a mean neutralizing titer of 9000, though this was not significantly higher than heterologous VS+RS immunized mice (mean titer of 7240) (Figure 2D). Homologous VS+VS vaccinated mice had the lowest mean neutralizing titers (1280) and the greatest variability in neutralizing titers (Figure 2D). Interestingly, heterologous VS+RS immunization had a 31-fold increase in neutralizing titers upon boost, higher than the 23- and 11-fold increases seen in RS+RS and VS+VS regimens, respectively. These data suggest that homologous mRNA and heterologous immunizations yield the best neutralizing responses of all conditions tested. Neutralizing titers measured by 50% focus reduction neutralizing test (FRNT_50_) were positively correlated with total IgG titers (Figure 2E). Taken together, these data show that all vaccination strategies elicit strong antibody responses. A homologous RS+RS induced the highest responses. However, heterologous boost induced the greatest increase in neutralizing antibody titers.

### 3.3. Homologous mRNA and Heterologous Immunizations Stimulate Robust Type 1 CD4+ and CD8+ Responses

To assess differences in T-cell responses induced by each vaccination strategy, mouse spleens were collected at 7 days post single dose vaccination or 7 days post boost (day 28 from experiment initiation) (Figure 2A). Splenocytes were harvested and stimulated overnight with a peptide pool covering SFTSV Gn or SFTSV Gc. Cells were analyzed by flow cytometry for the type 1 cytokines interferon γ (IFNγ), tumor necrosis factor α (TNFα), or interleukin 2 (IL-2) as well as for degranulation markers CD107a and Granzyme B (GranzB). Flow cytometry data were analyzed following the gating strategy shown in Figure 3A. In single dose immunization, only RS vaccinated animals showed strong CD8+ T-cell activation with both the Gn and Gc peptide pools (Figure 3B). Mice receiving a single dose of VS also showed increases in type 1 cytokines above negative controls RL and VE, though cytokine positivity was seen only in cells stimulated with the Gc peptide pool (Figure 3B). Stimulated CD8+ cells in both RS and VS mice showed a large increase in IFNγ+ cells and modest increases in all other cytokines including polyfunctional cells positive for multiple cytokines (Figure 3B).

In CD4+ T-cells, single dose immunization only induced a detectable response in animals immunized with RS (Figure 3D). CD4+ T-cell activation in RS immunized mice occurred with both the Gn and Gc peptide pools. Interestingly, higher activation of CD4+ T-cells occurred with the Gn peptide pool as opposed to CD8+ T-cells, which were more potently activated by the Gc pool. (Figure 3B,D). Activated CD4+ T-cells showed increases in all cytokines tested and particularly large increases in IFNγ and IL-2 (Figure 3D). Taken together, this data indicates that in single dose immunizations, RS induces much more potent cellular immune responses than VS. In prime-boost studies, CD8+ T-cell responses in RS+RS and VS+RS were relatively comparable and were far superior to a homologous VS+VS strategy. Animals immunized with RS+RS and VS+RS both had potent CD8+ immune responses when stimulated with Gn and Gc peptide pools (Figure 3C). Responses to Gn peptides were of smaller magnitude to Gc stimulation consistent with observations of single dose immunized mice. Interestingly, RS+RS and VS+RS achieved a similar response with stimulation by the Gc peptide pool, with approximately 42% of cells being cytokine positive. In contrast, VS+RS mice had a significantly lower response compared to RS+RS when stimulated with the Gn peptide pool (Figure 3C). As seen in mice receiving a single vaccine dose, cytokine positivity is primarily attributed to the robust production of IFNγ and a large CD8+ T-cell population that is IFNγ+TNFα+IL2- (Figure 3C). Animals receiving a homologous VS+VS immunization regimen did not achieve a detectable CD8+ T-cell response with Gn peptide stimulation and only achieved a mild response over negative controls when stimulated with Gc peptides (Figure 3C).

In contrast to CD8+ T-cells, cytokine profiling of CD4+ T-cells revealed a stronger response in heterologous VS+RS immunization than homologous RS+RS treated mice, though this was not statistically significant (Figure 3E). CD4+ responses were detected with both Gn and Gc peptide pools and were stronger in cells stimulated with Gn peptides, consistent with single dose immunization data (Figure 3D,E). Higher induction of CD4+ T-cells in heterologous immunized mice is surprising due to no detection of CD4+ T-cell responses with either peptide pool in mice receiving a single dose VS immunization (Figure 3D,E). Perhaps differing vaccination strategies lead to different tissue homing for activated T-cells explaining the weak T-cell responses seen in VS and VS+VS immunized mice. CD4+ responses had a cytokine profile featuring high levels of both IFNγ and IL-2 as well as a large population of triple positive cells producing IFNγ, TNFα, and IL-2 (Figure 3E). All trends shown here are consistent with analysis of splenocytes 14 days post immunization (Appendix A) and when cells are stimulated for only 6 h, though magnitudes of responses are decreased.

To determine CD8+ T-cell functionality, we also assessed positivity to the cytotoxic marker GranzB and the degranulation marker CD107a. In animals receiving a single dose immunization, only RS induced CD107a+GranzB+ cells when stimulated with both the Gn and Gc peptide pool (Figure 3F). Animals receiving the VS vaccine appeared to have a very slight but statistically insignificant increase in CD107a+GranzB+ CD8+ T-cells stimulated with the Gc peptide pool (Figure 3F). In animals receiving a boost, RS+RS and VS+RS mice induced equivalent levels of CD107a+GranzB+ CD8+ T-cells when stimulated with Gn peptides (Figure 3G). When stimulated with Gc peptides, VS+RS induced significantly higher levels of CD107a+GranzB+ CD8+ T-cells than the homologous RS+RS strategy (Figure 3G). For mice immunized with VS+VS, no significant induction of CD107a+GranzB+ CD8+ T-cells occurred (Figure 3G). Taken together, the T-cell analysis presented here suggests that in single dose immunizations, RS is superior. When a boost is administered, heterologous VS+RS and homologous RS+RS vaccination regimens perform equivalently overall.

### 3.4. Antibody Isotype Analysis Indicates mRNA Induced Some Type 2-Associated Immunity Despite Lack of Type 2 Cytokines in Stimulated T Cells

To determine whether immunization strategies skewed responses towards type 1 or 2 immune profiles, we performed SFTSV Gn/Gc ELISAs to quantify IgG1 vs. IgG2c levels. In C57BL/6 mice, IgG2c is associated with a type 1 immune response with IFNγ driving class switching to IgG2c [115,116]. In contrast, IgG1 is associated with a type 2 immune response and isotype switch to IgG1 is driven by IL-4 [115,116]. In single dose immunizations, VS immunization drives primarily an IgG2c response with only low levels of IgG1, as expected from a virus known to drive a type 1 response (Figure 4A). Surprisingly, immunization with RS led to approximately equal levels of IgG1 and IgG2c indicating a balanced type 1 and type 2 response (Figure 4A). Despite a mixed response, RS and VS immunized animals had similar titers of IgG2c (Figure 4A). Upon boost, VS+VS immunized animals had increased levels of both IgG1 and IgG2c though IgG2c titers remained approximately 2 logs higher than IgG1 (Figure 4A). Mice receiving the RS+RS regimen had a slight boost to IgG1 though isotype IgG2c showed the biggest boost in titer (Figure 4A). RS+RS mice had the highest mean IgG1 titer of any group by approximately 1 log (Figure 4A). Finally, heterologous VS+RS vaccination resulted in both IgG1 and IgG2c responses being boosted by approximately 1.5 logs. Heterologous VS+RS immunization had an intermediate polarization compared to homologous RS+RS and VS+VS regimens (Figure 4A).

To further examine immune polarization to type 1 or type 2 responses, we assessed peptide stimulated T-cells for the presence of the type 2 associated cytokines IL-4 and IL-5 at 7 days post immunization (Figure 4B). Surprisingly, despite the presence of the IL-4 driven IgG1 antibody isotype in some vaccination groups, no IL-4+ CD4+ T-cells were detected in any vaccination regimen (Figure 4C,D). Similarly, another type 2 cytokine IL-5 was not detected in any immunization regimen (Figure 4E,F). Type 2 cytokines continued to be undetectable when splenocytes were analyzed 14 days post immunization (Appendix A).

### 3.5. Prime-Boost Vaccination Regimens Are Fully Protective Against SFTSV Challenge

After characterizing the immune responses elicited by our various vaccination strategies, we next sought to determine whether these vaccines would provide protection from challenge with a pathogenic dose of SFTSV. Date of challenge was set as day 0, initial immunization occurred on days −35 for animals receiving two vaccine doses, and day −14 for animals receiving a single dose as well as animals receiving a boost. On day −1 and day 2 mice received α-IFNAR antibodies administered IP. Fourteen four-week-old C57BL/6 mice were initially included in each group, with 10 mice in each group evaluated for protection against pathogenicity and 4 mice in each group sacrificed for evaluation of SFTSV viral titers.

Animals receiving a single dose of either RS or VS lost no weight upon challenge and were protected by immunization (Figure 5A,B). A single mouse receiving VS immunization succumbed on day 1 post challenge as a result of injury during the virus challenge (Figure 5B). Negative control immunized mice receiving RL or VE were susceptible to severe pathogenicity and had severe weight loss after challenge and high lethality in RL immunized mice (Figure 5A,B). In prime-boost regimens, mice immunized with RS+RS, VS+RS, and VS+VS were all fully protected from challenges and exhibited no weight loss (Figure 5A,B). As with single dose immunized mice, RL+RL and VE+RL mice had severe weight loss and RL+RL mice succumbed by day 6 post-challenge with a 30% survival rate (Figure 5A,B). Animals receiving a VE prime appeared to be partially protected with an 80% survival rate, but again exhibited severe pathology (Figure 5A,B). Decreased lethality relative to what was seen in Figure 1 is hypothesized to be partially attributed to mice being older with a more mature immune system and more acclimated to the animal facilities due to a 35-day immunization protocol, as well as long lasting antiviral responses elicited by initial VE immunization [93,117].

Two days prior to SFTSV challenge, sera were taken from immunized mice for neutralizing antibody analysis. As previously reported, both single dose and prime-boost immunized animals had high titers of neutralizing antibodies (Figure 5C). In VS immunization group, two animals failed to sero-convert, yet both animals survived challenge, implying non-neutralizing antibodies and/or cellular responses mediated protection from pathogenic challenge with SFTSV (Figure 5C).

Lethal infections of SFTSV in humans are characterized by high viremia. To determine the ability of our vaccine candidates to control SFTSV infection, 4 mice from each immunization group were sacrificed at 4 days post challenge to determine viral titers in the liver, spleen, and serum. In single dose immunized animals, both RS and VS immunization decreased tissue SFTSV titers equivalently (Figure 5D). Similarly, all three prime-boost regimens, RS+RS, VS+RS, and VS+VS, induced significant decreases in SFTSV titers in the liver and spleen (Figure 5D). Significant decreases in titers in the serum were only observed in RS+RS and VS+VS immunized mice, though VS+RS immunization did decrease serum titers relative to RL+RL (Figure 5D). Immunization with VE also led to significant decreases in SFTSV titers in liver and spleen when given in single dose (Figure 5D). In prime-boost studies, VE+RL mice exhibited decreased SFTSV titers though these decreases were not significant and were of lower magnitude than in single dose animals (Figure 5D).

### 3.6. High Levels of Antibody Are Maintained for Months After Vaccination

An important factor in considering the efficacy of vaccines is the longevity of the immune response. To assess the longevity of the responses induced by our vaccines, animals were immunized as described in Figure 2A and maintained with sera collections occurring approximately every one to two months for one year. Serum analysis by FRNT_50_ and ELISA showed that immunized mice had similar titers as previously described at days 21 and 35 post immunization (Figure 2C,D and Figure 6A,B). Over time, animals receiving a single dose VS immunization had the lowest titers of all groups which stabilized at titers of approximately 100 and 2500 for FRNT_50_ and total IgG, respectively (Figure 6A,B). Animals receiving a single dose RS immunization and the prime-boost VS+VS regimen reached and maintained identical titers for both neutralizing antibodies and total IgG (Figure 6A,B). For animals receiving the RS+RS or the heterologous VS+RS immunization regimens, titers peaked at day 35 at similar levels but VS+RS animals had decreasing neutralizing titers while RS+RS animals maintained neutralizing antibody titers of approximately 6000 for 200 days before decreasing (Figure 6A). For VS+RS receiving animals, neutralizing titers began to plateau at approximately 180 days post initial vaccination with neutralizing titers of 1100 (Figure 6A). Interestingly, total IgG for both RS+RS and VS+RS had similar titers and a slight downward trend over time despite VS+RS having a more substantial decrease in neutralizing titers over the same period (Figure 6A,B). Taken together, all vaccines induce long lived antibody responses that plateau in the long term with neutralizing antibody titers that are likely protective.

## 4. Discussion

SFTSV is a recently emerged pathogen with a high case fatality ratio [1,12]. The expanding range of the vector responsible for transmitting SFTSV and the ability of the virus to spread in nosocomial and veterinary settings highlights the need for development of targeted vaccines and therapeutics to this pathogen [20,21,22,23,24,25,26,27,28,29,30,31,32,33]. In light of these factors, several health organizations including the NIAID and WHO have listed SFTSV as a priority pathogen for the development of vaccines and therapeutics [34,35]. This has led to the exploration of several vaccine platforms such as protein subunit, DNA, mRNA, live attenuated virus, whole inactivated virus, and recombinant virus vectored platforms [59,60,61,62,63,64,65,66,67,68,69,70,71,72,73]. In this study, we characterize and compare rVSV and mRNA immunizations in single dose and prime-boost regimens including a heterologous model with an rVSV prime and mRNA boost. The reverse regimen was not considered due to the likelihood of mRNA induced antibodies neutralizing the rVSV vaccine boost. Heterologous vaccination has recently come to the forefront of vaccine research due to the approval of heterologous “mix-and-match” use of the SARS-CoV-2 vaccines around the world [101]. This vaccine regimen offers the potential to combine the strongest characteristics of each vaccine platform used to impart the strongest possible immune response. It also sheds light on pragmatic decisions that some may face when making vaccination selections based on availability during outbreak situations. Our data suggest that heterologous immunization against SFTSV induces potent immune responses.

Most SFTSV vaccine studies have thus far used *Ifnar*^−/−^ mice as a challenge model to evaluate vaccine efficacy [59,60,63,68,69,70,72]. While these models have been sufficient and yielded data suggestive of vaccine efficacy, the model is not suitable for all studies due to its immunocompromised status. IFN signaling is essential for innate immune functions and for the development of adaptive immune responses [113,114,118]. Following a previous study that indicated an antibody blockade against IFNAR made wild-type mice susceptible to lethal SFTSV infection, we were able to utilize this novel model to immunize immunocompetent mice for characterization of immune responses [58]. Utilizing this α-IFNAR blockade, these mice could then be made susceptible to SFTSV infection for vaccine efficacy studies. Currently, the only other immunocompetent animals available that exhibit lethal disease upon SFTSV challenge are cats and ferrets [55,56]. While both animal models are superior to the α-IFNAR blockade in the sense of not requiring immunomodulation, they have many drawbacks. Both cats and ferrets are more expensive to purchase and maintain, and both animals have increased likelihoods of injuring researchers with possible zoonotic transmission [30,119]. Additionally, ferrets only manifest lethal disease when aged >4 years, which make them useful in some scientific contexts but difficult to reliably stock for experiments [55]. Finally, there is a lack of available reagents and tools for the analysis of immune responses in both cats and ferrets, precluding detailed immunological phenotyping as presented in this work.

Work from our group has previously demonstrated that rVSV-SFTSV vaccines are safe and efficacious. Using *Ifnar*^−/−^ mice, we previously demonstrated that rVSV platforms induce robust titers of neutralizing antibody titers. Furthermore, these responses were protective from lethal challenge with SFTSV and even provided protection when passively transferred prophylactically or therapeutically to naïve mice then lethally challenged with SFTSV [69]. Work on mRNA vaccines against SFTSV have demonstrated that vaccines targeting the Gn domain of the SFTSV glycoprotein elicit powerful antibody and cellular responses which confer protection from lethal challenges. Additionally, these studies demonstrated that mRNA vaccinated animals maintain high titers of IgG antibodies and neutralizing antibodies for up to 15 weeks [61,63]. In this study, we build on prior work by using an mRNA vaccine encoding both SFTSV glycoprotein subunits and combining mRNA and rVSV based vaccines in a prime-boost regimen. In agreement with prior work, we demonstrate that all vaccine modalities evaluated produce high total IgG and neutralizing antibody titers. Additionally, we demonstrate that these titers are maintained for almost a year after initial immunization. We and others have previously shown, by passive transfer studies, that antibodies are a correlate of protection against SFTSV [48,49,50,51,59,68,69,72,120]. The high levels of antibodies maintained over time thus suggest these vaccination regimens are likely to provide long lived immunity from SFTSV infection.

T-cell responses and their contributions to protection from SFTSV remain largely unexplored. One study using vaccinia virus immunizations has shown that CD8+ T-cell depletion does not impact survival upon lethal challenge [72]. This data, however, is confounding as depletion was performed in mice that were immunized and thus had SFTSV directed antibody responses [72]. Despite weak neutralizing titers, total IgG titers were high and could mediate protection through opsonization, complement activation, and natural killer cell mediated antibody dependent cellular cytotoxicity [72]. Indeed, CD8+ cell depletion did not affect survival in mice immunized with a vaccinia strain encoding SFTSV Gn/Gc but did have an impact in mice receiving just SFTSV N as an antigen [72]. Since antibodies cannot bind to N, which is only found within cells or virions, this suggests that, excluding antibody mediated protection, T-cells do in fact contribute to protection [72]. Other groups using DNA vaccine platforms have also demonstrated that animals are protected when immunized with internal SFTSV proteins which do not induce neutralizing antibodies [59]. These studies suggest that T-cells may be important in the control and clearance of SFTSV upon infection. One study using a SFTSV Gn encoding mRNA vaccine analyzed CD8+ and CD4+ T-cell responses. The authors found modest activation of both T-cell populations when analyzed 14 days post vaccination. This study suggested that mRNA vaccines induce good cellular responses, though the cytokine profile and granzyme B activity in vaccinated mice was not evaluated [63]. Despite these prior studies, cellular responses induced by SFTSV vaccines remain poorly characterized. Here, we showed that mRNA vaccination as a single dose or in a prime-boost model potently induces polyfunctional CD4+ and CD8+ T-cells. CD8+ responses in heterologous vaccinated animals are comparable to homologous mRNA immunization though heterologous vaccinated animals exhibit higher degranulation of Granzyme B in cells stimulated with the SFTSV Gc peptide pool. Interestingly, heterologous immunization consistently activates more CD4+ T-cells than homologous mRNA vaccination, though these differences are not statistically significant. Surprisingly, immunization with rVSV-SFTSV in a single dose or prime-boost regimen induced weak or no T-cell responses. This is perhaps due to rVSV-SFTSV being considerably attenuated by switching the cognate glycoprotein to SFTSV Gn/Gc, which is not abundant on the cell surface where VSV buds [68,69,121]. This attenuation likely results in weak in vivo viral replication which is further controlled by VSV’s sensitivity to IFN. Taken together, heterologous and homologous mRNA immunizations induced the strongest T-cell responses, which other studies suggest may be a contributing factor in protection from SFTSV infections.

In designing vaccines, it is important to consider the polarization of immune responses towards a type 1 or 2 response [85]. Previous studies have shown that improper immune polarization may have deleterious impacts on the immunized patient [122]. This is best demonstrated in respiratory syncytial virus studies that determined the tested vaccines drove a type 2 response resulting in vaccine associated enhancement of disease [122]. Viral diseases and other intracellular pathogens are best controlled by type 1 responses characterized by cytokines such as type 1 IFNs, IFNγ, TNFα, and IL-2 along with antibody isotypes IgG2a/c (depending on mouse strain) [85]. In contrast, type 2 responses are suited for large extracellular pathogens and produce IL-4, IL-5, and IL-13 along with antibody isotypes IgE and IgG1 [85]. In antibody analysis of type 1 and 2 responses, we found that mRNA immunization led to a mixed polarization with approximately equal titers of IgG1 and IgG2c (Figure 4A). Upon boost with mRNA, these animals saw increased titers in both isotypes. In contrast, immunization with the rVSV platform led to a primarily IgG2c response. Heterologous immunization resulted in a mixed phenotype with high levels of IgG2c and intermediate IgG1 titers. Interestingly, cellular analysis indicated a strong type 1 phenotype in mRNA immunized animals (for single and prime-boost regimens) and heterologous immunized mice with no IL-4 and IL-5 being detected. Perhaps tissues that were not sampled, such as the lymph nodes, may contain IL-4+ CD4+ T-cells or timing for the detection of these cytokines is crucial in our vaccination strategies. This might also be explained by mRNA vaccines being potent activators of T follicular helper cells, which have been previously shown to be driven to a mixed polarization despite strong type 1 polarization in other CD4+ T-cells [123,124]. Further characterization of T-cell subsets and lymphoid T-cells is necessary to ascertain the reasons for a mixed polarization in antibody analysis versus a strong type 1 skewing in analyzed T-cells.

Despite differences in immune profiling, all tested conditions were protective upon SFTSV challenge. Interestingly our negative control mice immunized with VE and VE+RL exhibited severe illness characterized by weight loss but were partially protected from death caused by the SFTSV challenge. This is presumably from activation and long-lasting effects of interferon stimulated genes upon immunization with VE. Similar observations of long-lasting non-specific vaccine responses have been reported after vaccination with the live attenuated polio virus vaccine and the live attenuated pertussis vaccine [125,126,127]. Other studies have implied that long-lasting non-specific immunity may be due to long-lived IFN responses, likely due to upregulation of IFN stimulated genes, inhibiting subsequent infection [117]. Infections with VSV induce strong IFN responses which are known to activate pathways that make an animal refractory to subsequent infection. This has previously not been described in other studies using similar rVSV negative controls, likely due to the use of *Ifnar*^−/−^ mice which cannot initiate the innate immune responses that wild-type mice can. Despite this protection in a negative control, we can still conclude that our VS immunizations induce specific protective responses not due to innate immunity as no weight loss was seen in VS or VS+VS immunized animals, in contrast VE and VE+RL immunized animals did survive challenge but suffered severe weight loss. Additionally, a temporal effect can be observed in tissue SFTSV titers in animals receiving a single dose VE immunization as opposed to VE+RL animals. Challenge of VE immunized animals occurred 14 days post immunization, in these animals SFTSV tissue titer was similar to titers in mice receiving RS or VS immunizations. In prime-boost VE+RL animals, challenge occurred 35 days after the last RE dose was delivered. In these animals, SFTSV tissue titers were decreased relative to our RL+RL negative control but not as low as animals receiving RS+RS, VS+RS, or VS+VS. This is consistent with the hypothesis of IFN mediated protection as we would expect waning protection over time.

Overall, the data presented highlight the potential for both mRNA and rVSV based vaccines against SFTSV. In homologous immunization, the mRNA platform greatly outperforms rVSV in immunologic stimulation. However, immune characterization cannot currently be directly correlated to protection. Due to this, extensive work is required to determine whether the differences seen in immunologic responses will translate to clinical differences. Additionally, other factors need to be considered when choosing a vaccine platform. The mRNA platform has been limited by cold chain requirements, which are difficult to maintain in isolated rural areas. However, ongoing research has contributed to the development of more stable mRNA vaccines that may be stored at 4 °C or even room temperature, allowing for greater distribution of these vaccines globally [98,128,129,130,131,132]. Meanwhile, the rVSV-platform is well proven in isolated rural areas as demonstrated by the effective rollout of the rVSV based EBOV vaccine [79]. An additional attribute of rVSV vaccines, that may be important in the context of an outbreak, is that they have been shown capable of providing protection when administered immediately pre- or even post-exposure [91,92,93,94,95,96,97]. Indeed, rapid protection may be related to the partial protective effects that we observed in the VE immunized mice. Whether this is the case for rVSV-SFTSV should be explored. As has been suggested by other studies, heterologous immunization induces a somewhat superior or equivalent cellular response to homologous mRNA vaccination, and a somewhat weaker or equivalent humoral response [101,102,103,104]. These data are supported by human and animal studies of SARS-CoV-2 [101,102,103,104]. Approval of vaccines designed to be delivered in heterologous strategies faces regulatory hurdles due to the need to demonstrate safety for two platforms. However, as we have recently seen in the SARS-CoV-2 pandemic, the use of heterologous immunizations has great potential for use in emergency situations where availability of different vaccine platforms may be lacking. Currently there are no approved SFTSV vaccines, in this study we demonstrate two viable platforms for potential development and three highly efficacious prime-boost regimens.

## 5. Conclusions

In this work, we provide an analysis of prime-boost vaccination regimens using rVSV-SFTSV and mRNA. Our data are consistent with previous reports concerning the immunogenicity of rVSV-SFTSV and mRNA with a single immunization. We conclude that a prime-boost regimen using mRNA encoding SFTSV Gn/Gc induces superior neutralizing antibody titers and T cell responses. Importantly, we also conclude that an rVSV-SFTSV prime followed by an mRNA boost is highly immunogenic and likely to be protective against severe disease.

## Figures and Tables

**Figure 1 viruses-17-01095-f001:**
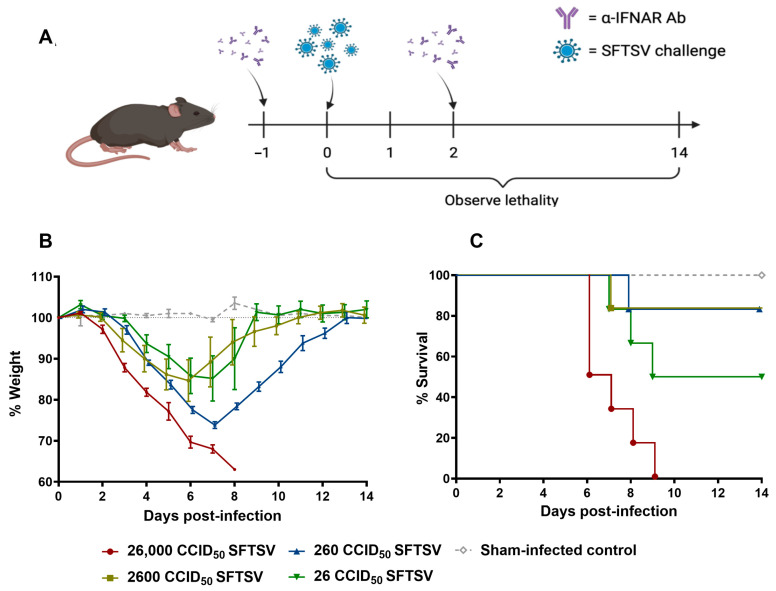
Determination of SFTSV LD_50_ in mice treated with α-IFNAR blockade. (**A**) Scheme for LD_50_ studies. (**B**) Weight loss and (**C**) survival in C57BL/6 mice challenged with 26, 260, 2600, or 26,000 CCID_50_ of SFTSV following α-IFNAR blockade. Panel A created using BioRender.com.

**Figure 2 viruses-17-01095-f002:**
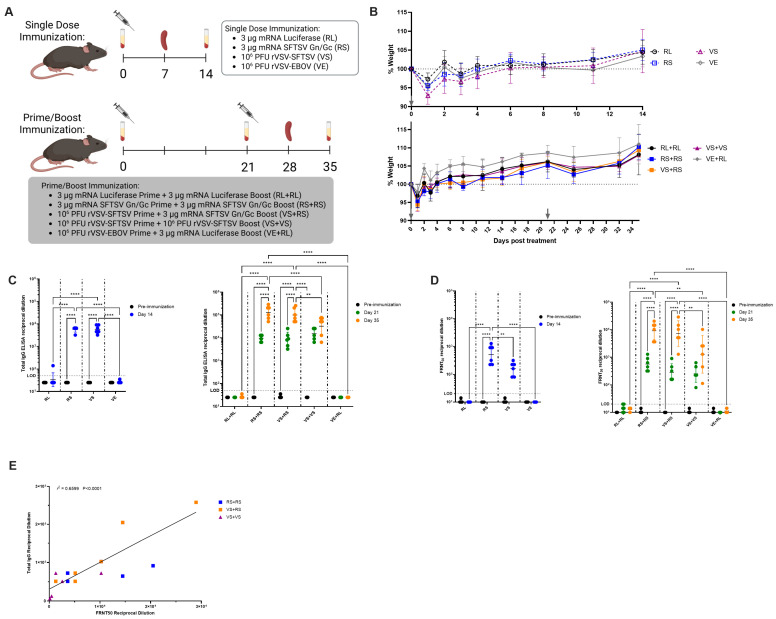
Immunization with mRNA and rVSV-SFTSV platforms induces potent humoral responses. (**A**) Study schematic. Serum samples were used to assess antibody responses via FRNT_50_ and ELISA. Spleen samples were used for T cell analysis via peptide stimulation and intracellular cytokine staining. (**B**) Weight loss in mice immunized in single dose (**top panel**) or prime-boost regimens (**bottom panel**). Gray arrows indicate dates of immunization. (**C**) Total IgG titers as determined by ELISAs in single dose (**left panel**) and prime-boost regimens (**right panel**). (**D**) Analysis of neutralizing antibodies by FRNT_50_ in single dose (**left panel**) and prime-boost regimens (**right panel**). (**E**) Total IgG and FRNT_50_ titers plotted against each other from mice receiving prime-boost immunizations at day 35. Simple linear regression *r*^2^ = 0.6599. Each panel includes data from 5 to 6 mice per group with equally split sexes. Limit of detection (LoD) of 20 and 50 for FRNT_50_s and ELISAs, respectively, are represented by a dashed line. Two-way ANOVA with Tukey’s multiple comparison test used in panels (**C**,**D**) (*p* = ** < 0.0021, **** < 0.0001). Panel A created using BioRender.com.

**Figure 3 viruses-17-01095-f003:**
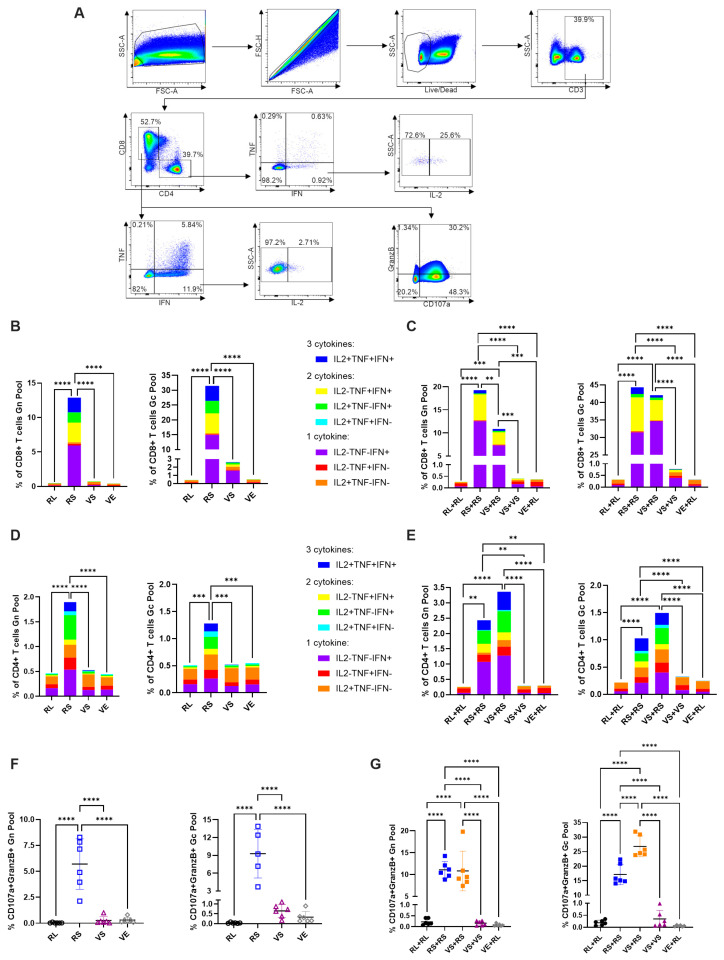
Single dose RS and prime-boost regimens RS+RS and VS+RS induce potent cellular responses. (**A**) General gating strategy used for analysis of flow cytometry data from splenocytes stimulated overnight with SFTSV Gn or Gc peptide pools 7 days after immunization. (**B**) Cytokine profiling of CD8+ T-cells in single dose immunized animals and (**C**) prime-boost immunized animals. (**D**) CD4+ T-cell cytokine profiling in single dose immunized animals and (**E**) prime-boost immunized animals. (**F**) Analysis of degranulation in CD8+ T-cells by CD107a and Granzyme B are shown in single dose immunized animals and (**G**) prime-boost immunized animals. Throughout figure, Gn peptide pool stimulated cells are shown on the left and Gc peptide pool stimulated cells are shown on the right. Each panel includes data from 5 to 6 mice per group with equally split sexes. Ordinary one-way ANOVAs with Tukey’s multiple comparisons test were used in panels (**B**–**G**) (*p* = ** < 0.0021, *** < 0.0002, **** < 0.0001). In panels (**B**–**E**) analysis was run on total cytokine positive cell percentage, not based on specific cytokine profiles.

**Figure 4 viruses-17-01095-f004:**
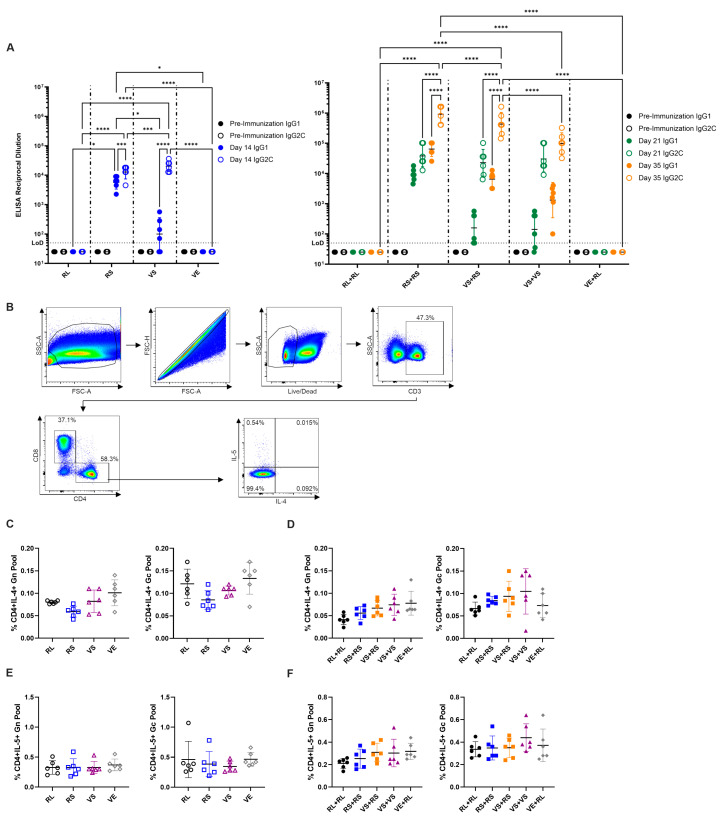
Immunization with mRNA and rVSV platforms does not induce type 2 cytokines IL-4 and IL-5 despite IgG1 type 2 response. (**A**) Analysis of mouse sera in single dose (**left panel**) and prime-boost regimens (**right panel**) for antibody serotypes IgG1 and IgG2c by ELISA. Limit of detection (LoD) of 50 for ELISAs is represented by a dashed line. (**B**) General gating strategy used for analysis of flow cytometry data from splenocytes stimulated overnight with SFTSV Gn or Gc peptide pools 7 days after immunization. (**C**) IL-4 cytokine profiling of CD4+ T-cells in single dose immunized animals and (**D**) prime-boost immunized animals. (**E**) CD4+ T-cell IL-5 cytokine profiling in single dose immunized animals (**F**) and prime-boost immunized animals. Each panel includes data from 6 mice per group with equally split sexes. Two-way ANOVA with Tukey’s multiple comparison test used in panel A (*p* = * < 0.0332, *** < 0.0002, **** < 0.0001). Ordinary one-way ANOVAs with Tukey’s multiple comparisons test used in panels (**C**–**F**), no significant comparisons were found. Throughout (**C**–**F**), Gn peptide pool stimulated cells are shown on the left and Gc peptide pool stimulated cells are shown on the right.

**Figure 5 viruses-17-01095-f005:**
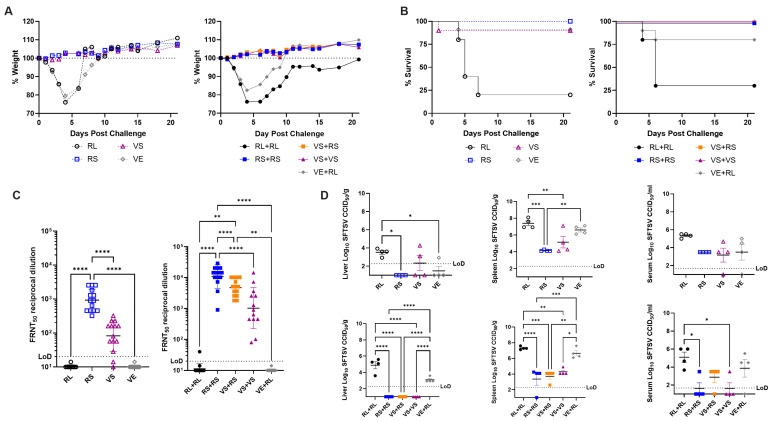
All immunization conditions are protective from pathogenic SFTSV challenge. C57BL/6 mice were immunized according to previously described vaccination regimens. Mice received α-IFNAR antibody blockades on day −1 and 2 post challenge with challenge (day 0) occurring 14 days after the last received immunization dose. (**A**) Average weight loss and (**B**) survival are shown for both single dose and prime-boosted animals. (**C**) Serum was collected from all animals 2 days prior to challenge for neutralizing antibody titer determination. (**D**) SFTSV titers in liver, spleen, and serum were determined in a subset of mice sacrificed at 4 days post SFTSV challenge. Ordinary one-way ANOVAs with Tukey′s multiple comparisons test was used in panels (**C**,**D**) (*p* = * < 0.0332, ** < 0.0021, *** < 0.0002, **** < 0.0001).

**Figure 6 viruses-17-01095-f006:**
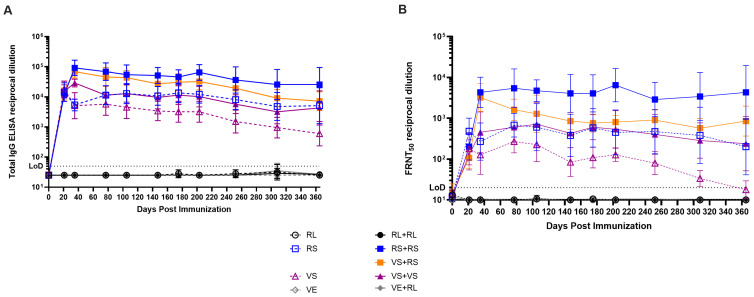
All immunization conditions elicit long lived antibody responses. (**A**) Total IgG titers as determined by ELISA in immunized mice. (**B**) Analysis of neutralizing antibodies by FRNT_50_ in immunized mice over extended period. Each panel represents data from 5 to 6 mice per group with equal sex distribution. Limit of detection (LoD) of 20 and 50 for FRNT_50_s and ELISAs, respectively, are represented by a dashed line.

## Data Availability

The original contributions presented in this study are included in the article/Appendix A. Further inquiries can be directed to the corresponding author.

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
