# Peer review of "Homologous and Heterologous Vaccination Regimens with mRNA and rVSV Platforms Induce Potent Immune Responses Against SFTSV Glycoprotein"

_viruses, 2025, doi:10.3390/v17081095_

Round 1
Reviewer 1 Report
Comments and Suggestions for Authors
SFTS is an important endemic and zoonotic infectious disease in Asia. Authors evaluated the efficacy of mRNA-based vaccine and rVSV-SFTSV in vivo. They presented the platforms and the prime-boost vaccine design. These data are essential to develop vaccines of SFTSV. However, there are some points to correct.
- Page 4, Line 157-160 : Please refer to the approval number of animal experiments.
- Page 4, Line 160 : Please describe the humane endpoints in this study. What points are set as the humane endpoints?
- Page 7, Line 281-295 : Please mention the limit of detection in antibody neutralization assay.
- Page 10, Line 380-381 : Authors said that the antibody class switching was thought to be continued since the titer of IgG increased. In that time, IgG1, IgG2c, or other class antibodies were not measured yet. I think it is not enough to say that just because of the increase of IgG titer.
- Page 12, Line 452-455 : Authors said that a heterologous vaccination strategy using an mRNA prime and rVSV boost was not considered due to rVSV-SFTSV containing SFTSV Gn/Gc on the virion surface which would likely be neutralized by antibodies elicited by an mRNA vaccination, thereby preventing any significant boost. In the VS+VS immunized group, the humoral immune response and cytotoxic immune response was not induced highly compared to those of other groups. I think the VS+VS vaccination strategy was similar to the RS+VS vaccination strategy. Did authors boost with VS because authors expected that cellular immune response would increase or be induced after VS boost following VS vaccination? I think it would be good to discuss this point as well in the results or discussion.
- Page 13, Line 465-475 : Authors said that RS is superior to induce T-cell. A single dose of VS induced the lowest neutralizing antibodies and T cell responses. In case of VS, is the concentration or titer of VS related to these results? If mice are immunized with higher titer of VS, are there some changes of results? Did the authors investigate this? If so, please discuss that point in the discussion section.
- Page 15, Line 498-500 : Why were the type 1 cytokine not assessed?
- Page 15, Line 505-507 : It might be better to place this sentence in the discussion section.
- Page 17, Line 529-532 : SFTSV is associated with a high fatality rate in older adults. Although the situation could be different between SFTSV-infected mice and SFTS patients, it is difficult to relate the decreased lethality with older mice.
- Page 18, Line 555-558 : It is better to add to the period of sera collection.
- Negative control vaccine, RL and VE+RL, immunized mice had an increased survival rate after the SFTSV challenge. I think this is valuable data, but this also poses a limitation in this study. “VE immunized mice appeared to be protected from lethal challenge with 90% of mice surviving, despite severe weight loss.” I’m wondering the protection efficacy of the prime-boost of VE-VE vaccination. Do authors have data about this? And, why did authors use VE vaccination as negative control not rVSV (empty vector)? It is better to describe these points in the discussion section.
- Figure 2A : It is confusing. It is better to describe as other formats, such as a bar line of experiment schedule.
- Figure 2C, 2D : The title of Y axis could be changed with “Total IgG titer (ELISA)” and “VN titer (FRNT50)”.
- Figure 5A : In contrast to figure 2B, there are no error bars. Also, entirely, the number of mice used in this study was not mentioned. It is necessary to add these points.
- Figures : Some asterisks for the significances are above the same line. It is confusing and unclear between which groups the significant differences exists. It is necessary to separate the bar under the asterisks.
Reviewer 2 Report
Comments and Suggestions for Authors
This study explored relevant vaccine platforms for SFTSV comparing different vaccine platforms about single dose, homologous, and heterologous prime-boost regimens in mice. The work addresses a significant unmet need for SFTSV countermeasures and offers data on immunogenicity, protection, and immune polarization. However, the fundamental limitations of the α-IFNAR blockade model for immunogenicity studies, the unresolved contradiction in immune polarization data, the lack of specificity in protection prompted by negative controls, and the incomplete exploration of the heterologous strategy significantly diminish the reliability, novelty, and impact of the findings.Therefore, I recommend publish after revision.
Comments: 1. The description in lines 40 is incorrect, “countries” must be corrected into “countries and regions”. Taiwan isn’t a country.
2. The α-IFNAR blockade model was used in this study but this treatment may effect adaptive immune responses which may lead interference to the results of this experiment. The authors have not adequately demonstrated the rationality of this model. Adaptive immune responses are critical in vaccine research, the observed immune responses may lead to anomalous outcomes or fail to represent responses in a fully immunocompetent host because of the α-IFNAR blockade. This may cause significant doubt on the reliability and translational relevance of all immunogenicity data. The authors should provide more detailed reasons for selecting this model.
3. The significant protection has been observed in negative control groups. Although the authors attribute this to non-specific IFN-mediated protection, this may makes it difficult to determine the true specific contribution of the SFTSV antigens to protection. The authors should undertake further research on this aspect. 4. The negative control was inappropriately designed. When the authors explored prime-boost regimens they only set VE+RL and RL+RL as the negative control groups and they didn’t give enough reasons to explain the rationality of this protocol. The VE+RL group is incapable of attaining the study endpoints for this prime-boost regimens study, because in single dose study the VE group has already demonstrated a significant protective effect. Considering the significant protective effects observed in the VE group, the absence of a VE+VE control group may undermines the reliability of their conclusions. The authors should conduct additional experiments or provide justification for their research.
5. The authors believed RS induced significant IgG1 (associated with a type 2 immune response) but no detectable IL-4 or IL-5 from splenocytes is a major unresolved discrepancy. The authors speculate about unexamined lymph nodes or timing but provide no experimental evidence to support these possibilities. Because it’s very important in vaccine research, the authors should do more work to explain this contradictory.
6. In heterologous regimens research the authors choose VS+RS group and give up RS+VS group for the reason of predicted neutralization. To ensure the integrity of the study, experiment is needed to fully evaluate heterologous potential or the authors should provide a more compelling and detailed explanation.
7. The analysis of experimental results for heterologous regimens is methodologically unsound. The conclusions "induce superior immune responses" or "slightly superior" are overstated based on the presented data such as minor T cell differences and waning neutralizing antibodies which is insufficient to support the findings. It is recommended to conduct further research on this aspect or implement revisions accordingly.
8. Regarding article content, it is recommended that in Introduction section the mechanisms of mRNA and rVSV vaccines should be briefly elaborated to enhance the completeness of this article.
Round 2
Reviewer 1 Report
Comments and Suggestions for Authors
Authors have revised carefully. I understood very well. However, I’d want authors to correct three points.
